# Gynaecological Health Patterns and Motherhood Experiences of Female Professional Football Players

**DOI:** 10.3390/ijerph22020136

**Published:** 2025-01-21

**Authors:** Dimakatso Althea Ramagole, Dina Christa Janse van Rensburg, Charlotte Cowie, Ritan Mehta, Gopika Ramkilawon, Babette M. Pluim, Gino Kerkhoffs, Vincent Gouttebarge

**Affiliations:** 1Department of Orthopedic Surgery and Sports Medicine, Amsterdam UMC Location University of Amsterdam, Meibergdreef 9, 1105AZ Amsterdam, The Netherlands; maki.ramagole@up.ac.za (D.A.R.); b.pluim@knltb.nl (B.M.P.); g.m.kerkhoffs@amsterdamumc.nl (G.K.); 2Section Sports Medicine, Faculty of Health Sciences, University of Pretoria, Pretoria 0028, South Africa; christa.jansevanrensburg@up.ac.za; 3The Football Association (England), National Football Centre, St George’s Park, Needwood DE13 4PD, UK; charlotte.cowie@thefa.com (C.C.); ritan.mehta@thefa.com (R.M.); 4Department of Statistics, Faculty of Natural and Agricultural Sciences, University of Pretoria, Pretoria 0028, South Africa; gopika.ramkilawon@up.ac.za; 5Academic Center for Evidence-Based Sports Medicine (ACES), 1105AZ Amsterdam, The Netherlands; 6Amsterdam Movement Sciences, Aging & Vitality, Musculoskeletal Health, Sports, 1105AZ Amsterdam, The Netherlands; 7Amsterdam Collaboration on Health & Safety in Sports (ACHSS), IOC Research Center of Excellence, 1105AZ Amsterdam, The Netherlands; 8Football Players Worldwide (FIFPRO), 2132LR Hoofddorp, The Netherlands

**Keywords:** female, professional, football, menstrual cycle, contraceptives, motherhood

## Abstract

The aim of this paper is to explore the gynaecological health patterns, contraceptive use, body perception, and motherhood experiences of female professional football players. The participants were recruited via email using FIFPRO (Football Players Worldwide). Online questionnaires were completed by consenting participants. The mean age at menarche was 13.5 years with an average cycle length of 26 days, and a bleeding period of 5 days. Cycle irregularities were experienced by 30%, and menstrual symptoms by 74%. Half of the participants used contraceptives, 60% using hormonal contraceptives, primarily oral contraceptive pills (38%), followed by implants (20%). The body satisfaction score was normal but there was a high drive-for-thinness (DT) score. The motherhood rate was low (1%), with participants experiencing normal conception, vaginal delivery, return to training after 6 weeks, and return to competition after 12 weeks. Our findings are consistent with findings in other elite female athletes with cycle irregularities and a significant number of cycle-related symptoms. The majority of those using contraceptives preferred hormonal contraceptives, reflecting trends seen in other elite athletes. While body satisfaction scores were normal, there was a high DT score, similar to that observed in lean and weight-category sports. The motherhood rate was low, consistent with previous findings in professional football players and other elite athletes. This may be due to a lack of financial support during pregnancy and the post-partum period. FIFPRO and its affiliated unions are negotiating better contracts for female football players.

## 1. Introduction

In recent years, there has been an increase in female participation in sports. Female physiology has been investigated in different elite female athletes, but there remains a gap in research on active female professional football players. Differences in male and female physiology have been well documented, but training protocols for females remain lacking. Fluctuations in female sex hormones, progesterone and oestrogen may affect sports performance across different menstrual cycle phases, which usually lasts for 28 days but may vary between 21 and 25 days [1]. The use of hormonal contraceptives (HC) downregulates endogenous hormone production because they provide supplementation for the hormones and might affect exercise performance negatively as compared to naturally menstruating females [2].

A systematic review of injuries in male and female youth football showed that the overall incidence rate was higher for females (5.70 injuries/1000 h in males and 6.77 injuries/1000 h in females). Match injury incidence was higher than training injury in both groups (14.43 injuries/1000 h in males and 14.97 injuries/1000 h in females) [3]. Males tend to sustain more muscle/tendon injuries, whilst females sustain more joint/ligament injuries. These findings indicate sex differences in injury rates and anatomical location. Different injury prevention strategies should be implemented [3].

Wing et al. also found that male football players ran approximately twice the distance of female players. This may have been due to the time played by males, which was longer (20 min per quarter) compared to 15 min per quarter for females. Still, high-speed running distances were higher in males than females [4].

Although data are still scarce in women’s football, mental health symptoms ranging from substance abuse and sports-related distress to anxiety have been documented and have shown an increase in elite players ranging from 1% to 3% for substance abuse, and up to 65% for psychological distress over a period of 12 months [5]. These trends are a cause for concern and monitoring female athletes is recommended.

Several surveys have investigated contraceptive use, including the methods chosen and the reasons behind their use. Clarke et al. [6] reported that amongst 177 participants from three different football codes (rugby league, rugby union/sevens, Australian football), a third (*n* = 58) were on HC, and 47 (26%) were using the oral contraceptive pill (OCP). Seventy-one percent indicated that contraceptive use was for pregnancy avoidance, 38% to regulate the menstrual cycle and 36% to manage menstrual pain. Baumgartner et al. [7] reported amongst 408 athletes from 92 different sports where 43.4% participated in lean sports that primary amenorrhoea was predominantly common in athletes with a body mass index (BMI) lower than 21.7 kg/m^(2)^_._ They reported that 25.8% of the athletes were on the OCP, and 54.4% of athletes were not using contraceptives at all or using non-HC [7].

Motherhood experiences (ME) have unique challenges for female athletes, as highlighted in the article “Pushing for change” [8] where 25 athletes were interviewed. In a similar article, ‘We are not superhuman, we’re human’ [9], 18 athletes were interviewed [9]. Some concerns raised by these athletes included ‘training a new body’ postpartum; safe return to training; breastfeeding and exercise; and managing time between sports and motherhood. These athletes emphasised the need to develop best practices and secure funding for a safe return to play (RTP) [5,6]. Elite female swimmers and basketball players also tended to end their athletic careers earlier than their male counterparts due to lower financial income, maternity pressure and wanting to start a different professional career [10].

Choosing between motherhood and elite sports participation requires careful pregnancy and fertility planning [11]. Challenges regarding training during pregnancy cited by elite athletes include safety, fatigue, physical discomforts, family responsibility, cultural beliefs, lack of information [12], and the lack of a supportive network and funding during the pre-and post-natal periods [11]. Dietz et al. [13] identified 11 challenges among 16 elite athletes from different sports from Austria and Germany. These were categorised into organisational/environmental (family support, time management and travelling) financial (financial insecurity and contracts), personal (reaching set goals in sports, finding the right partner, doubts about being a mother and keeping independence) and physical factors (performance and health after pregnancy) [13].

The American College of Obstetricians and Gynaecologists (ACOG) Committee advises that women who participate in vigorous exercise should be encouraged to continue with exercise during pregnancy and the postpartum period, and those who are sedentary should initiate safe physical activity practices. The benefits of exercise include a decrease in gestational diabetes, caesarean and operative vaginal deliveries, and a shorter post-partum recovery time [14].

Male athletes, on the other hand, can often balance parenthood with sports without disrupting their careers [11]. Smith et al. [15] identified three different themes amongst 10 elite father-athletes viz, (1) fatherhood improves athletic performance, (2) fatherhood impedes performance and (3) athlete fathers achieve a trade-off between sports and parenting [15].

Gender disparities in salaries and player contracts have been a concern amongst elite female athletes. FIFPRO reported that median net salaries for men worldwide ranged between USD 1000 and USD 2000, with average contract periods of 2 years [16]. Women earn an average monthly salary of USD 600, with 50% getting no pay from their clubs and almost 50% with no employment contracts. The contracts are averaged at 12 months [17].

FIFPRO has started efforts globally to improve working conditions and negotiate contracts for professional female football players and has successfully negotiated 14 weeks of paid maternity leave from 2021. They also launched a 5-step guide for return-to-play after childbirth, which includes a period of 0–6 weeks of return to early exercise, a 5–7 week health check, 12 weeks of return to exercise, followed by football-specific training, and finally return to high-performance [18].

Based on the considerations mentioned above, this study aimed to achieve two primary objectives, which are to explore gynaecological health patterns in female professional football players, including the menstrual cycle, menstrual symptoms, contraceptive use and body perception, and to examine motherhood experiences, including methods of conception and delivery, and return-to-play/return to competition after childbirth.

## 2. Materials and Methods

### 2.1. Study Designs and Ethical Considerations

An observational prospective cohort study over a follow-up period of six months was conducted using the Strengthening the Reporting of Observational Studies in Epidemiology statement in order to guarantee the quality of reporting [19]. The Medical Ethics Review Committee of the Amsterdam University Medical Centers (Amsterdam UMC, location AMC) provided ethical approval for the study (Drake Football Study: NL69852.018.19|W19_171#B202169). The study was conducted in accordance with the Declaration of Helsinki (2013).

### 2.2. Participant Selection

The study population consisted of female professional footballers recruited by FIFPRO and affiliated national unions. Inclusion criteria were: (a) being a professional footballer, (b) being female, (c) being able to read and comprehend texts in English or French. In our study, the definition for a professional footballer was that she (i) trains to improve performance, (ii) competes in the highest or second highest national league, and (iii) has football training and competition as a major activity (way of living) or focus of personal interest, devoting several hours in all or most of the days for these activities, and exceeding the time allocated to other types of professional or leisure activities. Sample size calculation was calculated using the formula $n=\frac{Z^{2}P\left(1,-,P\right)}{d^{2}}$ where *n* = sample size, *Z* = statistic for a level of confidence (1.96 for 95% confidence level), *P* = Expected prevalence or proportion, and *d* = Precision [20]. This indicated that at least 40 participants were needed to reach a power of 80% (confidence interval (CI) of 95% and absolute precision of 7%) under the assumption of an anticipated population proportion (prevalence) of 5%. Recruitment of participants by FIFPRO and its union affiliates was carried out via email from September 2020 to May 2021 leading to enrolment of 80 players, and 74 completed the questionnaires. Data collection for this study on this cohort was conducted in 2021 and 2022.

### 2.3. Dependent Variables

Various female-specific variables were collected through single questions related to (history of) pregnancy, fertility (e.g., in vitro fertilisation, sperm donation), miscarriage, adoption, date of menarche, menstrual cycles, (history of) amenorrhoea, use of contraceptive measures (e.g., oral, implant), experiences related to motherhood (e.g., vaginal delivery, caesarean delivery), return to training (RTT)/return to competition (RTC) and body perception.

Body perception was measured by using the body dissatisfaction (BD) and drive for thinness (DT) scores. These were measured with two of the 12 primary scales of the Eating Disorder Inventory (EDI-3) [21]. BD relates to the disapproval of the overall shape of one’s body and the size of specific body regions of particular concern. DT consists of perceptual, behavioural, and attitudinal parts and is probably triggered by weight status misperception, and may lead to the development of disordered eating [22].

### 2.4. Descriptive Variables

Participant characteristics and several descriptive variables (e.g., age, height, body mass index (BMI), player position, and stress fractures) were collected using an electronic questionnaire.

### 2.5. Procedures and Data Storage

A baseline and a follow-up electronic questionnaire were set up in English and French (CastorEDC, CIWIT B.V, Amsterdam, The Netherlands), including all study outcome measures. In addition, the following descriptive variables were added to the baseline questionnaire: level of education, parallel activity (e.g., study, work), level of football, number of seasons as a professional footballer, history of hospitalisation, history of eating disorders, and smoking status. (See Appendix A). Information about the study was emailed to potential participants by FIFPRO and affiliated national unions, and procedures were hidden from the principal researcher for privacy reasons. If interested in the study, all participants gave informed consent and received access to the baseline questionnaire. The follow-up questionnaire was sent six months later. Each questionnaire took about 15 min to complete. The responses to the questionnaires were coded and made anonymous for reasons of privacy and confidentiality. The electronic questionnaires were saved automatically on a secured electronic server that only the principal researcher could access. Players participated voluntarily in the study and were not rewarded for their participation.

### 2.6. Statistical Analyses

The Statistical software, R (version 4.2.1; http://www.r-project.org, access date 23 June 2022) was used to perform all data analyses. Descriptive analyses (mean, standard deviation (SD), frequency and proportions) were performed for all variables included in the study, including Body Mass Index (BMI) calculated as a ratio of weight/height squared [23]. This software was used to analyse gynaecological health patterns and motherhood experiences. All statistical tests were performed at a 5% level of significance.

## 3. Results

### 3.1. Participant Characteristics

Seventy-four women athletes were recruited, with a mean age of 24.95 (95% CI: 24.32, years and a median (IQR) of 25.00 (23.00, 27.00). The mean BMI was 22.14 (95% CI: 21.8, 22.48) and a median (IQR) of 22.02 (21.13, 23.10). The number of years as a professional player was 5.74 (95% CI: 5.03, 6.45). Table 1 depicts demographics, football characteristics, career level and employment status. The majority of our participants were defenders *n* = 25 (34%), followed by forwards *n* = 22 (30%), midfielders *n* = 17(23%) and goalkeepers *n* = 10 (14%), as depicted in Figure 1. The level of football is depicted in Figure 2.

### 3.2. Gynaecological Health Patterns and Body Perception

Table 2 depicts gynaecological health patterns, BD and DT scores. The mean age at menarche was 13.5 (1.3) years, with a menstrual cycle of 26 (13.09) days and duration of menstrual bleeding of 5 (3.42) days. Most participants, *n* = 55 (74%), reported pain before or during the bleeding period. Half of the participants, *n* = 37 (50%), used contraception, mostly hormonal contraceptives (HC), predominantly the oral contraceptive pill (OCP) *n* = 14 (38%), and implants *n* = 7 (20%). Participants had a normal body perception score of 4.88 (5.08), which is comparable to their BMI. They had a higher-than-expected DT score of 3.59 (5.04), which may be attributed to ethnical diversity.

The menstrual cycle is depicted in Figure 3, and menstrual-related pain in Figure 4.

### 3.3. Motherhood Experiences and RTT/RTC

Only 1 (1.4%) of the 74 participants had two children, both conceived naturally and delivered vaginally. Her RTT was at 6 weeks and RTC at 12 weeks on both occasions.

A report on 69 retired elite women footballers in 18 different countries showed that 54 (78.3%) were not on any contraception, 17 (24.6%) were mothers with 1–3 children, 51 (73.9%) had never been pregnant. The majority (86.7%) who fell pregnant did so within 2 years of the desire to fall pregnant. Only four of these participants (5.8%) fell pregnant during their football career, and the mean RTC after childbirth was 22 weeks [24]. These findings confirm the general trend of contraception use and low motherhood rates during the active career of female professional footballers.

## 4. Discussion

The findings from this study provide a baseline of active female professional football players’ health behaviours, whilst previous studies reported on retired elite athletes [25], or elite athletes from other sports [13,15]. Our findings were that: (1) cycle irregularities were common, (2) most participants experienced cycle-related pain, (3) hormonal contraceptives, particularly the OCP, were the most commonly used, and (4) motherhood experiences were rare.

### 4.1. Menstrual Cycle

Menarche is the average age at which menstrual bleeding starts in young females, and usually, this is between the ages of 12 and 13 years. Our participants’ mean menarche age of 13.5 (1.3) years is within normal limits. Our results showed that 30% of our participants had cycle irregularities, which is higher than Swiss elite athletes from various sports, where approximately 15% reported such issues [4]. However, our results are lower than those for Danish elite athletes [26] and the United Kingdom [27], where 51% of athletes reported cycle irregularities. Retired semi- and professional football players reported a 40% prevalence of menstrual irregularities, with 22% experiencing amenorrhoea lasting longer than 3 months [25].

Menstrual pain was reported by 74% of our participants. Symptoms were present before the bleeding period in 20% and during the menstrual phase in (54%) participants. This is comparable to the findings in a UK survey where 77% of elite athletes reported menstrual pain [28]. Elite rugby players reported menstrual pain in 93% of participants [28], and 84% of elite Danish athletes reported negative symptoms related to their menstrual cycle [26].

Several studies have shown that athletes believe that their performance declines when they experience menstrual-related symptoms [26,28,29,30], and symptoms include higher perceived fatigue during this period [31]. Our study did not investigate the impact of the menstrual cycle on exercise performance, highlighting the need for future research to address this gap in the literature on female professional football players.

### 4.2. Contraceptive Use

Our findings confirm that 50% of our participants were on contraceptives, mainly HCs (60%), predominantly the OCP (38%), followed by implants (20%) and injectable contraception (2%). These findings are consistent with findings amongst other elite athletes, with 50% on HCs, predominantly the OCP (68%), followed by the implants (13%), injection (4%) and intra-uterine devices (3%) [6]. In elite Australian football codes, HC use was 33%, with 81% on the OCP [7]. A survey of Swiss players reported that 46% were on contraception, and 26% used the OCP [7], and a study on Danish players reported 57% use of HCs and all of them on OCP [26]. Amongst retired professional football players from the United States (US), the trend was similar, although more participants (64.5%) had used HCs, predominantly the OCP (94.5%) [25]. Some of our participants reported the use of more than one contraceptive method. Our study was not designed to investigate reasons for the use of different methods and what ‘other’ methods were used; however, we hypothesise that these may include the ‘cycle method’ where intercourse is avoided during ovulation. Further research is warranted on this topic.

### 4.3. BD and DT

Our participants had a normal BD score of 5. This finding aligns with findings in federated Spanish amateur football players with a normal BD core of 5 [32]. We are not aware of other similar studies on female professional football players. A systematic review and meta-analysis across different sports and genders highlighted the importance of sports participation in fostering positive body image. The study found that female athletes had higher BD scores than males, whilst normal-weight athletes had higher BD scores than underweight ones, and compared to non-athletes, female athletes tend to have a higher body image perception [33].

Our participants had a high mean DT of 4, which was not expected in football. Higher DT scores are common in aesthetic sports (e.g., gymnastics) and weight-category sports (e.g., lightweight rowing), where leanness gives a competitive advantage [34,35]. We are not aware of other comparable studies on female professional football players. We acknowledge the ethnical diversity of our participants, which may affect our findings.

High DT has been correlated to susceptibility to musculoskeletal injuries due to low energy availability in the National Collegiate Athletic Association (NCAA) Division II female athletes and higher time loss from training [36]. In this NCAA survey, 82% of participants sustained musculoskeletal injuries in-season. The mean number of injuries/training days loss was higher in the high DT group (over 2.0 ± 0.3), with 9.8 (2.6) days loss from training, whilst the low DT group (under 2.0 ± 0.3) had 6.9 (2.6) days loss [36]. Monitoring of our participants is necessary to prevent or minimise injuries.

We found that 70% of our participants did not have another paid job, 58% were not studying, and 13 (17.6%) smoked. These factors, combined with a high DT, warrant further monitoring of these athletes to promote well-being. As confirmed in one study, high DT is a risk factor for the development of ED, and personal risk factors for DT include perfectionism, low self-esteem in sports, unemployment, and not studying [37]. Although ED was found to be higher in non-athletes compared to elite female football players [38], the combination of risk factors for ED and susceptibility to musculoskeletal injuries warrants further monitoring and research.

### 4.4. Motherhood Experiences

In our study, only one participant (1%) reported having had two pregnancies, and she conceived naturally, delivered vaginally, and returned to training and competition 6- and 12-weeks post-partum, respectively. The US study reported that 50% of the participants reported trying to fall pregnant, with a 94% success rate, mostly by intra-uterine insemination (51%), medication (49%) and in vitro fertilisation (43%) [25]. Our sample is too small to compare. Still, we could assume that pregnancy during a professional football career remains challenging as players might be afraid to jeopardise their careers due to motherhood potentially.

It was shown that some professional athletes opt to choose between motherhood and sports [6,11], but similar to our study, other athletes have successfully fallen pregnant and managed to RTT/RTC earlier than non-athletes [39]. Although early RTT has been associated with an increased risk for stress fractures, high BD and high DT [39], this has not been confirmed in our study.

FIFPRO reported in 2017 that only 2% of female professional football players were mothers, and almost 45% expressed that they would have to leave the game early to start a family [17]. Our study’s low percentage of motherhood is comparable to the FIFPRO findings. The choice between motherhood and professional football was not canvassed in our study, but it will form a basis for future research.

### 4.5. Practical Implications

Gynaecological health and motherhood choices form an integral part of elite female athletes. The establishment of this cohort in professional football will assist in further research, which can improve the health of these athletes and possibly prolong their sporting careers. We assume that the high DT in our cohort is due to the ethnical diversity of the participants. This should, however, alert the medical team to monitor these athletes closely. This will also form a basis for global education on addressing female-specific challenges in sports.

### 4.6. Strengths and Limitations

Our strength is that we established a cohort of active female professional football players that will be used for further prospective studies over a period of 10 years. Our limitations are the small sample size, the ethnical diversity of our participants, and the lack of in-depth analysis of the menstrual cycle and coping mechanisms, reasons for choice of contraception and motherhood. Further research is needed to address these shortcomings.

## 5. Conclusions

Among 74 female professional football players, 30% had cycle irregularities, and 74% had cycle-related symptoms. Two-thirds of those on contraceptives were on HCs, with 38% on the OCP and 20% on implants. This trend aligns with most elite athletes’ gynaecological patterns. Our participants had a normal body satisfaction score (BD = 5) but an unexpectedly high DT score of 4, possibly due to ethnical diversity. We found a low motherhood rate, which aligns with a previous report on professional football players. More research to analyse menstrual symptoms and coping strategies, contraceptive use and motherhood planning is warranted in female professional football players to ensure a longer and safer sporting career.

## Figures and Tables

**Figure 1 ijerph-22-00136-f001:**
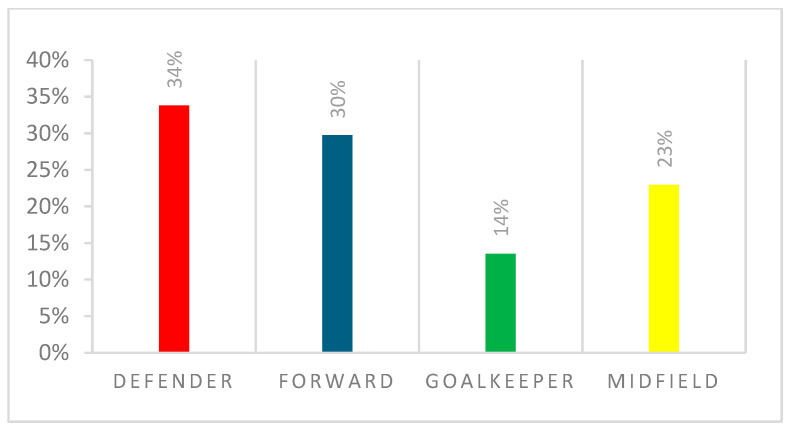
Field position.

**Figure 2 ijerph-22-00136-f002:**
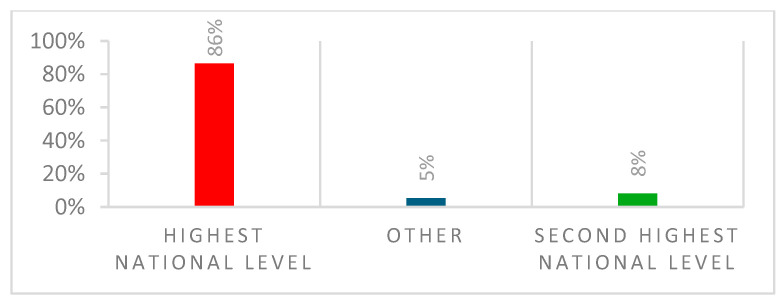
Level of football.

**Figure 3 ijerph-22-00136-f003:**
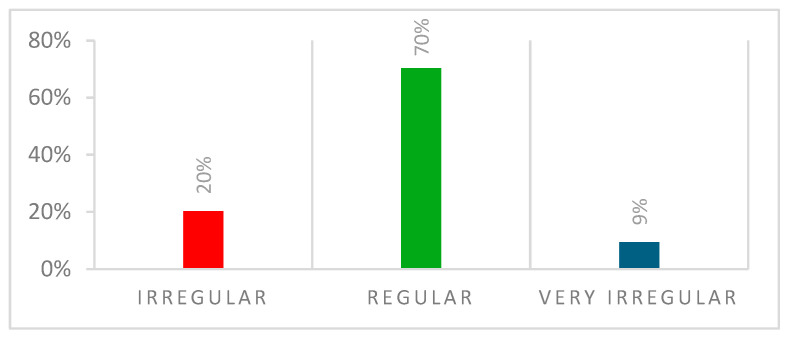
Menstrual cycle.

**Figure 4 ijerph-22-00136-f004:**
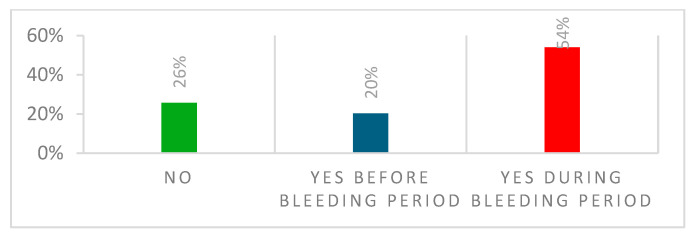
Menstrual-related pain.

**Table 1 ijerph-22-00136-t001:** Demographics, football characteristics, employment status and career level.

Demographics		All Participants (*n* = 74)
Age (yrs) mean (SD)		24.95 (2.69)
Height (cm) mean (SD)		168.39 (5.38)
Weight (kg) mean (SD)		62.81 (5.4)
BMI (kg/m^2^) mean (SD)		22.14 (1.47)
Stress fractures *n* (%)		11 (15)
Low bone density *n* (%)		1 (1.35)
Other characteristics		
Field Position *n* (%)	defender	25 (34%)
	forward	22 (30%)
	midfield	17 (23%)
	goalkeeper	10 (13%)
Number of seasons as a professional player Mean (SD)		5.74 (3.12)
Employed	yes	22 (30%)
Level of Football *n* (%)	highest national level	64 (87%)
	second highest national level	6 (8%)
	other	4 (5%)

Yrs = years; SD = standard deviation; cm = centimetres; kg = kilograms; BMI = Body Mass Index.

**Table 2 ijerph-22-00136-t002:** Gynaecological health patterns, BD and DT.

Variable	Characteristics	All Participants (*n* = 74)
Age of Menstruation (yrs) Mean (SD)		13.47 (1.3)
Duration of menstruation cycle (days) Mean (SD)		25.81 (13.09)
Duration of menstrual bleeding period (days) Mean (SD)		5.16 (3.42)
Description of menstruation cycle *n* (%)	regular	52 (70%)
	irregular	15 (20%)
	very irregular	7 (10%)
Pain during menstruation cycle *n* (%)	during bleeding period	40 (54%)
	none	19 (26%)
	before bleeding period	15 (20%)
	none	37 (50%)
	pill	14 (38%)
Contraception Methods *n* (%)	condom	12 (32%)
	other	8 (22%)
	implant	7 (20%)
	hormone replacement	4 (10%)
	injection	1 (2%)
Body dissatisfaction Mean (SD)		4.88 (5.08)
Drive for thinness Mean (SD)		3.59 (5.04)

SD = standard deviation; BD = Body dissatisfaction; DT = Drive for thinness.

## Data Availability

A baseline and a follow-up electronic questionnaire were set up in English and French (CastorEDC, CIWIT B.V, Amsterdam, The Netherlands), including all study outcome measures. The responses to the questionnaires were coded and made anonymous for reasons of privacy and confidentiality. The electronic questionnaires were saved automatically on a secured electronic server that only the principal researcher could access.

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
