# Peer review of "Gynaecological Health Patterns and Motherhood Experiences of Female Professional Football Players"

_ijerph, 2025, doi:10.3390/ijerph22020136_

Round 1

Reviewer 1 Report

Comments and Suggestions for Authors

Overall I enjoyed reading the manuscript and think that is has merit. Specific and general comments are below:

line 66 - explain more about the contracts that some of the players have signed. Can you provide a range of salaries, or averages? Of the roughly 50% who do not have a contract, how are they paid? Do any of the contracts have clauses about what happens if a player become pregnant, etc? When you write "comparable to those of their male counterparts," do you mean in salary only, or additional aspects. 

Are there any efforts to organize/unionize these women's leagues?

You touch on why BD was higher than you expected. Could you hypothesize why these scores were higher than you expected? Is there data that shows that most/many women in Netherlands have high BD, whether they play a sport or not?

I think that you could write more about the sports where female athletes have high BD. I believe that they are high in sports that are judged (gymnastics) rather than refereed (soccer). 

In America BD among females tends to differ between white and African-Americans. Were all of these women of the same ethnicities?

Author Response

Thanks for yor comments.

I have attended to the recommendations and attached a cover letter.

Reviewer 2 Report

Comments and Suggestions for Authors

I went over the manuscript. Although the authors made a considerable efforts to conduct the research project. I have some concerns need to be considered before publication.

Introduction: The introduction outlines relevant topics in female athlete health but requires restructuring and additional depth. While it touches on key issues (menstrual health, contraception, motherhood), it lacks sufficient context specific to professional football. The progression from general female athlete health to football-specific challenges needs better development.

Specific comments:

  1. Lines 39-43: Split the complex sentence: "In recent years... endurance athletes[1]"
  2. Lines 43-46:Add statistical context to early menarche claim
  3. Lines 47-50: Provide specific statistics from referenced surveys
  4. Lines 51-56: Restructure to highlight specific challenges. Add transition sentence linking ME to football specifically. Include statistics about ME in professional sports
  5. Lines 57-60: Include quantitative data on decreased risks
  6. Lines 61-65: Consolidate survey findings. Add percentages/statistics where available
  7. Lines 66-69: Compare with male professional contracts. Add specific financial implications. 
  8. Consider Adding a paragraph about current state of women's professional football and  specifically addressing research gaps
  9. Include section on physiological differences affecting female athletes

Materials and Methods: The methodology section lacks sufficient detail in key areas and requires better organization. While the basic study design is clear, critical methodological elements need elaboration to ensure reproducibility and validate the scientific rigor.

Specific comments:

  1. Specify exact time period of data collection. 
  2. Include power calculation for sample size justification.
  3. Clarify recruitment process and response rate in more details.
  4. Specify data security measures.

Results: The Results section is well-organized into three main subsections (participant characteristics, gynaecological health patterns/body perception, and motherhood experiences) but lacks depth in statistical analysis and data presentation.

Specific comments:

  1. Participant Characteristics: Add statistical measures of distribution (range, median) beyond just mean/SD. Include confidence intervals for key measurements. Add subgroup analyses by playing position. Consider presenting demographics in more detail
  2. Gynaecological Health Patterns and Body Perception: Add statistical comparisons between subgroups and Report p-values for significant findings. Include correlation analyses between variables (e.g., BMI vs BD scores). Add analysis of relationships between contraceptive use and cycle irregularities. Clarify "other" contraceptive methods category
  3. Motherhood Experiences (Lines 153-155): While limited by sample size (n=1), provide more detailed case information. Add comparative context with general population statistics. Include qualitative information about motherhood planning/intentions
  4. Present more data visually using figures/graphs

Discussion: The Discussion systematically addresses key findings but requires stronger integration of results and broader implications for female professional football. While organized by themes, it needs deeper analysis of practical applications and future directions.

Specific comments:

  1. Introduction Section: Add clear summary statement of major findings. Provide stronger context for uniqueness of study. Remove numbered format for smoother narrative flow
  2. Menstrual Cycle Section: Add discussion of performance implications. Include potential intervention strategies. Strengthen link between findings and practical recommendations
  3. Contraceptive Use Section: Add analysis of contraceptive choice rationale. Discuss implications for performance management. Address gaps in current knowledge.
  4. Body Dissatisfaction/Drive for Thinness Section: Strengthen analysis of unexpected DT findings. Add recommendations for monitoring/intervention. Include potential cultural and social influences.
  5. Motherhood Experiences Section: Include policy implications. Strengthen recommendations for support systems
  6. Add limitations section

Update the abstract and conclusion according to the improvement suggestions.

Author Response

Thanks for your comments and input.

The issues raised have been addressed, and I have attached a cover letter.

Round 2

Reviewer 2 Report

Comments and Suggestions for Authors

The revised manuscript shows significant improvement but requires minor refinement.

The introduction needs more specific information about female football players' health challenges. Consider adding recent statistics and explaining how playing football affects women's bodies differently from men's.

The methods section should better explain how you analyzed the data. Please clarify what happened to incomplete questionnaires and how you handled the follow-up questionnaire information.

The results would be stronger with more statistical analysis. Add confidence intervals for your measurements and show how different factors (like playing position or age) might be related to each other. Explain what "other" contraceptive methods means.

The discussion should better explain why you found high drive-for-thinness scores and what this means for women's football. Also explain how team doctors can use this information to help their players.

Author Response

Dear Reviewers

Thank you for your comments and suggestions. Comparisons and statistics between male and female players have been added. Other comments have been addressed in the cover letter.
